# Metronidazole for Treatment of *Clostridioides difficile* Infections in Brazil: A Single-Center Experience and Risk Factors for Mortality

**DOI:** 10.3390/antibiotics11091162

**Published:** 2022-08-28

**Authors:** Joana Darc Freitas Alves, Augusto Yamaguti, João Silva de Mendonça, Cristiano de Melo Gamba, Cibele Lefreve Fonseca, Daniela K. S. Paraskevopoulos, Alexandre Inacio de Paula, Nair Hosino, Silvia Figueiredo Costa, Thaís Guimarães

**Affiliations:** 1Infectious Diseases Department, Hospital do Servidor Público Estadual de São Paulo, São Paulo 04029-000, Brazil; 2Microbiology Department, Hospital do Servidor Público Estadual de São Paulo, São Paulo 04029-000, Brazil; 3Infectious Diseases Department, Hospital das Clínicas, University of São Paulo, São Paulo 05508-220, Brazil

**Keywords:** *Clostridioides difficile*, metronidazole, mortality

## Abstract

We describe the epidemiology of *C. difficile* infections (CDIs) focused on treatment and analyze the risk factors for mortality. This is a retrospective cohort study of CDI cases with a positive A/B toxin in the stool in 2017–2018. We analyzed the demographic data, comorbidities, previous use of antimicrobials, severity, and treatment, and we performed multivariate analysis to predict the 30-days mortality. We analyzed 84 patients, 37 (44%) of which were male, where the mean age was 68.1 years and 83 (99%) had comorbidities. The percentage of positivity of the A/B toxin was 11.6%, and the overall incidence density was 1.78/10,000 patient days. Among the patients, 65.4% had previous use of antimicrobials, with third-generation cephalosporins being the class most prescribed, and 22.6% of cases were severe. Treatment was prescribed for 70 (83.3%) patients, and there was no statistically significant difference between the initial treatment with metronidazole and vancomycin even in severe cases. The 30-day mortality was 7/84 (8.3%), and the risk factors associated with mortality was a severity score ≥2 (OR: 6.0; CI: 1.15–31.1; *p* = 0.03). In this cohort of CDI-affected patients with comorbidities and cancer, metronidazole was shown to be a good option for treating CDIs, and the severity score was the only independent risk factor for death.

## 1. Introduction

*Clostridioides difficile* has become an increasingly prevalent enteric pathogen both in communities and in healthcare-related infections, being mainly responsible for antimicrobial-associated diarrhea. *C. difficile* infections (CDIs) range from mild and self-limited diarrhea to pseudomembranous colitis and toxic megacolon, with the latter being the most severe form of the disease, which can result in death [1,2].

The high incidence and severity of CDIs result in part from the emergence of a hypervirulent strain of *C. difficile*, known as BI/NAP1/027. The BI/NAP1/027 strain has a gene that encodes the binary toxin in addition to toxins A and B and has a negative regulator of toxin production A and B, resulting in an increase in their expression [3]. Surveillance data from CDIs in the USA from 2005 to 2007 indicated that 43.0% (80/186) of *C. difficile* strains were positive for BI/NAP1/027 [4]. The most recent data from 2008 to 2013 showed consistent frequencies ranging from 24.7% to 28.4% [5]. This strain, although frequent in the USA, has not yet been described in Brazil [6].

In Brazil, published studies on CDIs are related to animal colonization, genotypic analysis, and sensitivity tests or diagnostic tests, with very few existing studies on the epidemiology of this infection. In a recent Brazilian study conducted in 5 hospitals, there was a 15% prevalence of CDIs, and the molecular analysis of 35 strains demonstrated 17 different sequential types without finding the hypervirulent strain nor resistance to either metronidazole or vancomycin [6,7].

Considering the scarcity of Brazilian studies about the epidemiology of CDIs and the importance of knowledge of this infection in the hospital environment, our study aims to describe the epidemiology of CDIs, focused on treatment and analyzing the risk factors for mortality at the Hospital do Servidor Público Estadual (HSPE) of São Paulo, Brazil.

## 2. Results

We evaluated 84 patients with positive A/B toxins from 1 January 2017 to 31 December 2018. The percentage of positivity was 11.6% (84/725) and over several months ranged from 0 to 28%. During this period, the overall incidence density (ID) of CDI was 1.78/10,000 patient-days (84/470.896 patient-days).

Of the patients, 37 (44%) were male with ages ranging from 1 to 93 years (mean of 68.1 years). Ninety-nine percent of the patients had comorbidities, with the most common comorbidity being systemic arterial hypertension (67.5%) followed by diabetes mellitus (37.3%), gastrointestinal tract diseases and cardiac diseases (25.3%), chronic kidney disease (14,4%), and stroke (8,4%). Cancer was present in 15/84 (17.8%) patients.

Previous use of antimicrobials within 90 days prior to the diagnosis of CDI was observed in 55 (65.4%) patients, 15 (27.3%) used only one antimicrobial, and the other 40 (72.7%) used 2 or more antimicrobials in the period. Third-generation cephalosporins were used in 24 (43.7%) cases, beta-lactams or beta-lactamase inhibitors in 17 (30.9%) cases, carbapenems and glycopeptides in 14 (25.5%) cases, quinolones in 11 (20%) cases, and clindamycin and metronidazole in 5 (9.1%) cases. The duration of previous antibiotic therapy ranged from 1 to 84 days, with a mean of 13.1 days and a median of 10 days.

Regarding the time between hospital admission and the diagnosis of CDI, we observed that 10 patients were outpatients and were not hospitalized. This time ranged from −5 to 97 days, with an average of 18 days and a median of 10 days.

The laboratory parameters evaluated are described in Table 1.

The Clinical Score Infection (CSI) was possible to calculate in 62 (73.8%) patients, with 14/62 (22.6%) having a score ≥2, and they could quickly be classified as severe CDIs.

Treatment for a CDI was prescribed for 70 (83.3%) patients, and 13 patients did not receive treatment for this diagnosis. The reason why these patients did not receive treatment was that 3 patients were not evaluated, and in 10 patients, the information was not available since they were outpatients, and prescription information was not available in the electronic medical records.

Of the treated patients, 65 (92.8%) received treatment only with metronidazole, 1 (1.4%) received metronidazole + oral vancomycin, and 4 (5.7%) received only oral vancomycin. Sixteen (22.8%) patients received more than one treatment due to relapses. Of these, 13 (18.6%) patients received a second treatment, with 7 (53.8%) receiving metronidazole and 6 (46.2%) receiving vancomycin orally, while 2 (2.9%) patients received a third treatment: one of them with metronidazole, another with oral vancomycin, and one (1.4%) patient requiring a fourth treatment with oral vancomycin. The duration of the first treatment ranged from 2 to 22 days, and the mean treatment time was 10.8 days. There were no episodes of recurrence.

Among the patients who had severe CDIs and received treatment (*n* = 12), 10 (83.3%) patients started treatment with metronidazole. Of the patients treated with metronidazole, the mean treatment time was 11 days, and 2 (20%) patients died, both within 27 days of diagnosis. Two (16.7%) patients with severe CDIs received vancomycin as the initial treatment, one of which was associated with metronidazole, and one (50%) patient died 23 days after diagnosis.

The 30-day mortality was 7/84 (8.3%), and we could observe that 4/7 (57.1%) had CSI scores ≥2. These four patients with severe CDIs were analyzed in relation to treatment, and of these, only three were treated: two with metronidazole and one with vancomycin. Patients who received metronidazole used it for 7 days and died 27 and 28 days after the diagnosis of a CDI. The patient who received vancomycin used it for 12 days and died within 23 days of the diagnosis of a CDI. One severe patient was not treated for their CDI and died 12 days after this diagnosis.

For the analysis of risk factors for mortality, we performed univariate and multivariate analysis, described in Table 2 and Table 3.

It was not possible to perform multivariate analysis of the variable prior use of antimicrobials because all patients who were evaluated until death used antimicrobials.

## 3. Discussion

In our study, we found 84 patients with CDIs, which corresponded to 11.6% positivity of the A/B toxin. There are scarce data about the positivity rate in international studies, but in Brazilian studies, this rate ranged from 8.3% to 31.8% [7,8,9].

Similarly, we found a general ID of 1.78/10,000 patient days. Usually, this rate is calculated for 10,000 patient days, but we found studies where this rate was calculated for 1000 patient days and others where it was calculated based on population density. We calculated the ID per 10,000 patient days in order to allow comparisons. Data from the USA show an ID from 6.36 to 7.03, and data from Europe show an ID from 1.67 to 3.14/10,000 patient days [10]. Our ID was significantly lower when compared to the ID in the EUA. We could hypothesize some explanations for our lower rate: (1) there was a little diagnostic suspicion of CDIs in our hospital, (2) even when suspected, diagnostic tests were not always requested to confirm the diagnosis, and (3) the ID in our hospital was really low. On the other hand, when we observed the Brazilian ID of the published studies, we found an ID calculated for 1000 patient days of 0.4/1000 patient days, and ours would be 0.17 if calculated in this denominator [11]. Few Brazilian studies have calculated the ID for 10,000 patient days, and we found ID values of up to 5.45/10,000 patient days but in a specific population of patients submitted to hematopoietic stem cell transplantation [12].

The mean age of our patients was 68.1 years, according to the profile of the population assisted at HSPE. It is noteworthy that advanced age is considered a factor of poor prognosis in many severity scores, and because we attended predominantly elderly people, we should suspect this infection more often in hospitalized patients with risk factors [13].

Ninety-nine percent of the hospitalized patients had comorbidities. The exception was one patient whose age was 1 year. We can expect a high frequency of comorbidities in elderly patients, but we found the presence of malignancies in 18.1% of the patients [14]. This diagnosis is important because it is also considered a marker of severity for CDIs.

Sixty-six percent of our patients had previous use of antimicrobials, which is consistent with the literature data that mentions up to 60% of cases of CDIs being related to previous antimicrobial use. Among the antimicrobials cited as risk factors in the literature are beta-lactams and beta-lactams combined with beta-lactamase inhibitors [15]. In our series, the use of third-generation cephalosporin was more prevalent, but beta-lactams combined with beta-lactamase inhibitors were present in 30.9% of cases. Carbapenems and glycopeptides were previously used in 25% of cases, and this may reflect the greater severity of any previous nosocomial infections of these patients. The mean duration of previous antimicrobial use was 13.1 days, with no prolonged antimicrobial exposure in this population. Although the duration was not related to the development of CDIs, we could notice that there was no prolonged exposure time to antimicrobials.

The laboratory parameters were not available for all patients. This is a limitation of retrospective studies based on medical records analysis, but we observed that the laboratory parameters that indicate severity were present in 11.4%, 18.8%, and 66.7% of patients regarding leukocytosis, increased creatinine, and hypoalbuminemia, respectively. This analysis allowed the calculation of the severity score, where we found 22.6% of severe patients. This score is important for estimating mortality. The literature data show mortality ranging from 45% to 75% for moderate to severe cases, but these data are American, where the prevalence of the hypervirulent strain BI/NAP1/027 is high [16]. In Brazil, there has been no report of the BI/NAP1/027 strain, so clinical and laboratory severity criteria may be important to estimating mortality and guiding therapy [6].

Although our study had a high prevalence of elderly people and 18% were cancer patients, only 22.6% of the patients were considered severe. It is worth mentioning that we could only calculate the severity scores for 73.8% of the patients.

Treatment was performed in 83.3% of the patients, with metronidazole being the first choice for 91.4% of them. Only 5.7% received vancomycin as the first treatment. Due to the fact that we do not have the oral formulation of vancomycin or hypervirulent strain BI/NAP1/027, metronidazole was the first choice of treatment, and vancomycin was reserved for relapses or recurrences. Metronidazole was prescribed without stratification of the severity of disease. Therefore, we correlated severity with therapeutic choice and observed 28.5% of the 30-day mortality with metronidazole and 14.2% with vancomycin in patients considered severe, with no statistically significant difference. Therefore, in our series, metronidazole is still a good therapeutic option even in severe cases. Again, the fact that we did not have circulation of the BI/NAP1/027 strain may have contributed to the favorable outcomes [6].

The 30-day mortality was 8.3%, and 22.6% of patients were considered severe. Mortality in the USA and Europe ranges from 4% to 75% [17,18]. Even though we do not have the BI/NAP1/027 strain, which is associated with higher mortality rates, our mortality rate may be considered low, and considering the severity of our patients, it is important to reinforce the importance of suspicion and early and appropriate treatment through these patients in order to reduce this percentage.

In the analysis of risk factors for 30-day mortality, we found previous use of antimicrobials, cancer, and a CSI ≥2 with statistical significance. Obviously, the severity score is strongly correlated with death, and the previous use of antimicrobials is a risk factor for the development of the disease. Both may also contribute to a higher mortality rate [19]. When we submitted these variables to the logistic regression model, only a CSI ≥ 2 remained an independent risk factor for mortality (OR = 6.00; CI 95% = 1.15–31.1; *p* = 0.03). Considering that the CSI score is easy and very feasible to apply because it only requires the analysis of cancer history, leukometry, serum creatinine, and the albumin dosage, we recommend that this score be applied to patients diagnosed with CDIs in order to institute early treatment and decrease mortality.

Our study has several limitations. It is a retrospective study of a single center that assists elderly patients, which can lead to bias. We also did not analyze other risk factors, such as the use of antacids and enteral nutrition, which could contribute to the development of diarrhea [20]. We also did not analyze the duration of diarrhea after the introduction of specific therapy to evaluate the therapeutic efficacy of the drugs, but we analyzed the mortality rate over 30 days.

Another limiting factor is that we did not have microbiological evaluation of the cases because the diagnostic method used in our study was the positivity of the A/B toxin and not the culture for microbiological analysis of the strains.

Despite these limitations, we believe that this is an epidemiological study that can bring a great contribution to understanding the evolution of CDI cases in Brazilian patients.

## 4. Methods

This is a retrospective study that was conducted through the analysis of medical records of patients positive for the A/B toxin in stool samples from 1 January 2017 to 31 December 2018. The cases of the study were carried out through information from the microbiology laboratory, which provided a list of cases positive for the A/B toxin collected in this period. This test had a sensitivity of 57–83% and a specificity of 99% [21].

The HSPE is a tertiary teaching hospital, with 823 beds and 77 beds for intensive care units. It also has clinical and surgical hospitalization wards of various specialties. The microbiology laboratory performed A/B toxin tests using an immunochromatographic assay method (CD Toxin A/B ECO Test-TR.0031^®^).

The inclusion criteria were patients with diarrhea (more than 3 episodes of pasty stool or liquid feces in 24 h) and positive for A/B toxins in the study period. Patients with duplicates of positive tests within 14 days were considered a single case. The recurrence of infection was defined as a new positive A/B toxin test after 90 days. A relapse of CDI was defined as the recurrence of symptoms within 2–8 weeks of successful treatment of the initial episode [15].

The percentage of positivity was calculated by finding the number of positive samples of toxin A/B in the numerator and the number of samples requested in the denominator, excluding duplicates.

Patients were followed up with during hospitalization until hospital discharge (discharge or death). Mortality was analyzed 30 days after the result of positive A/B toxin tests.

The variables studied were demographic data, the presence of comorbidities, previous use of antimicrobials and its duration until 90 days prior, hospital admission data, laboratory parameters for calculating the severity score, therapeutic measures, and evaluation of the clinical outcome (discharge or death) during the same hospitalization within 30 days. We evaluated the following laboratory parameters: leukocyte count, serum creatinine, and serum albumin up to 48 h before or 48 h after the diagnosis of a CDI. These laboratory parameters were researched to evaluate CDI severity stratification according to the Clinical Score Infection (CSI) criteria [22]. For the CSI, we considered leukocytes above 20,000 cells/mm^3^, creatinine levels above 1.5 times the upper limit of normality (1.4 mg/dL), and albumin level less than 3.0 mg/dL. To calculate the CSI score, one point was assigned for each laboratory parameter and one point if cancer was present. When the score was ≥2, the case was considered severe.

All information regarding the patients was stored in a database using Excel 5.0. The analysis of qualitative variables was performed using a chi-squared test or Fisher’s exact test. ANOVA was used to analyze the differences between the means of the quantitative variables. The potential factors related to mortality were compared by univariate analysis, and all factors identified as significant were submitted to multivariate analysis by the multiple logistic regression model. The independent variables were expressed through their risk ratio (“odds ratio” (OR)), and their respective confidence intervals (CIs) of 95% were estimated. All probabilities of significance presented were bilateral and performed considering a significance level of 0.05 or 5.0%. Statistical calculations were performed using EPI-INFO version 7.2.

## 5. Conclusions

A CDI is a less prevalent infection (1.78/10,000 patient days) that affects elderly patients with comorbidities and cancer and with previous use of antimicrobials. Twenty-three percent of these infections were considered severe. Metronidazole was the first therapeutic choice, and it did not contribute to a higher mortality rate. The 30-day mortality was 8.3%, and the severity score was the only independent risk factor for death.

## Figures and Tables

**Table 1 antibiotics-11-01162-t001:** Analysis of laboratory parameters of patients with CDIs.

Laboratory Parameter	Leukocytes (Cells/mm^3^) *n* = 70	Creatinine (mg/dL) *n* = 69	Albumin (mg/dL) *n* = 33
Range	1.140–37.970	0.2–10.2	1.7–4.6
Media	11.210	1.63	2.7
Median	9.145	1.0	2.7
Leukocytes > 20.000	8 (11.4%)		
Creatinine > 1.5 × ULN *		13 (18.8%)	
Albumin < 3.0 mg/dL			22 (66.7%)

* Upper limit of normality.

**Table 2 antibiotics-11-01162-t002:** Univariate analysis of risk factors for 30-day mortality in patients with CDIs.

Variable	Dead *n* = 7 *n*(%)	Alive *n* = 77 *n*(%)	*p*
Male	4 (57.1)	33 (42.9)	0.10
Age (media in years)	67.9	68.1	0.09
Comorbidities	7 (100)	76 (98.7)	0.91
Systemic arterial hypertension	6 (85.7)	50 (64.9)	0.41
Diabetes mellitus	3 (42.8)	28 (36.3)	0.70
Gastrointestinal tract diseases	2 (28.5)	19 (24.6)	1.0
Cardiac diseases	4 (57.1)	17 (22.0)	0.06
Chronic kidney disease	3 (42.8)	9 (11.6)	0.06
Stroke	2 (28.5)	5 (6.4)	0.10
Prior use of antimicrobials	7 (100)	48 (62.3)	0.04
Third-generation cephalosporin	6 (85.7)	18 (23.3)	0.01
Beta-lactams and beta-lactamase inhibitors	5 (71.4)	12 (15.5)	0.03
Carbapenems	5 (71.4)	9 (11.6)	0.01
Glycopeptides	4 (57.1)	10 (12.9)	0.01
Quinolones	5 (71.4)	6 (7.7)	0.02
Clindamycin	1 (14.2)	4 (5.1)	0.36
Metronidazole	3 (42.8)	2 (2.5)	0.03
Time to diagnosis (media in days)	17	17.8	0.99
Cancer	2 (28.6)	13 (16.9)	0.05
Initial treatment for CDI (n = 70)	6 (85.7)	64 (83.1)	0.66
Treatment with metronidazole	5 (71.4)	60 (77.9)	0.65
Treatment with vancomycin	1 (14.2)	4 (5.1)	0.36
Length of treatment (media in days)	9.5	10.9	0.49
Leukocytes > 20.000 cells/mm3	2 (28.6)	6 (7.8)	0.13
Creatinine > 1.5 × ULN *	3 (42.9)	10 (13)	0.07
Albumine < 3.0 mg/dL	2 (28.6)	20 (26)	0.59
CSI ≥ 2 (n = 14)	4 (57.1)	10 (13)	0.01
Initial treatment with metronidazole	2 (28.5)	8 (10.3)	0.19
Initial treatment with vancomycin	1 (14.2)	1 (1.3)	0.16

* Upper limit of normality.

**Table 3 antibiotics-11-01162-t003:** Multivariate analysis of risk factors for 30-day mortality of patients with CDIs.

Variable	OR	CI 95%	*p*
Cancer	1.93	0.33–11.1	0.45
CSI ≥ 2	6.00	1.15–31.1	0.03

## Data Availability

The data were generated as part of the routine work of the departments.

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
