# Peer review of "Metronidazole for Treatment of Clostridioides difficile Infections in Brazil: A Single-Center Experience and Risk Factors for Mortality"

_antibiotics, 2022, doi:10.3390/antibiotics11091162_

Round 1

Reviewer 1 Report

Reviewers comment

In the study titled ‘Metronidazole for treatment of Clostridioides difficile infections in Brazil: a single-center experience and risk factors for mortality' by the authors, Alves et al. have done a retrospective study based on the medical data of 84 patients with C. difficile infected (CDI) in Brazil. The medical records were from Jan/2017 to Dec/2018. The confirmation of infection was by detecting the A/B toxin by immunochromatographic assay and other clinical signs.  30-day mortality was 8.3%, and cancer and a severity score  ≥ 2  were some of the risk factors associated with mortality.

The single-center retrospective, in general, has its own limitations, which the authors have mentioned in the manuscript. However, the study does give some information on the epidemiology of  CDI.  The writing of the manuscript is to be improved. Here are a few suggestions which might increase the value of the manuscript.

Line 23-24. Not clear Consider rephrasing the sentence.  Whether the authors tried to find differences in mortality between with and without treatment of drugs?  

Line 100: Does it mean the treatment was for 5-6 days? Rephrase the sentence.

Line 229: anova to be corrected as ANOVA

Line 207: The authors may consider explaining the methodology of estimating the toxin briefly.

Line 240: ‘Metronidazole was the first therapeutic choice, and it does not contribute to higher mortality.

Not clear what exactly the authors meant by ‘mortality’.  Why Metronidazole should contribute to higher mortality?

Author Response

The comments and answers are attached 

Reviewer 2 Report

This is an interesting retrospective cohort study on CDI in patients in Brazil who were diagnosed with a positive C. difficile toxin in stool. I have some comments that could help improve the manuscript:

1.     Line 16-17: positive A/B toxin in stool

2.     Line 45: [OR 1,93; CI 0,33-11,1; p=0,45]. To my understanding, this is not a statistically significant variable and should be deleted from that point of the abstract section

3.     Line 32: increasingly prevalent maybe?

4.     Line 48: Just to clarify something important: I feel that the references here refer to patients with diarrhea, not admitted patients in general. This is important to note when you state that the prevalence is 15%, because one may misunderstand the denominator in this rate

5.     Line 65: Previously used antimicrobials could be presented in a table either in the manuscript or, maybe better, in the supplementary table to allow the reader to more easily understand what had been used by the affected patients

6.     Table 1: A parenthesis is missing at the right bottom

7.     CSI is defined in the methods section which is (probably according to the manuscript requirements) after the results section. Thus, the CSI is mentioned without having defined the abbreviation. Please define all abbreviations when first used

8.     Line 86: Please define what you mean 2nd, 3rd treatment, etc. Was it for the same episode, as a therapeutic failure? Or are you talking about relapse?

9.     Line 143-144: glycopeptides, not glycopeptites

10.  Even though the authors define what a recurrence is in the methods section, I did not see any data regarding recurrence in the results section. This could be quite important since the patients were treated with metronidazole, and I am still skeptical about the risk of relapse since the latest guidelines have totally misplaced metronidazole and propose the use of fidaxomicin or oral vancomycin in patients with CDI

11.  Sometimes, variables identified in the univariate analysis to have a p-value somewhat higher (for example p<0.1) are chosen to continue to the multivariate logistic regression analysis. The authors could think about expanding the analysis to see if there are other factors independently associated with mortality

12.  Line 242: These are not truly risk factors, as they were not found to be significant in the multivariate analysis. You could either delete that or say that they were associated, but the only one found to be independently associated with mortality was the severity score

13.  Line 188-189: I don’t understand the statement ‘’it is a retrospective study, of a single center that assists elderly patients with cancer’’. You showed that a minority of the patients in this study were diagnosed with cancer

Author Response

The comments and answers are attached

Round 2

Reviewer 2 Report

The manuscript has been improved during the revision process